

# Daidzein alleviates osteoporosis by promoting osteogenesis and angiogenesis coupling

Junjie Jia[1,2,3], Ruiyi He[1,2,3], Zilong Yao[1,2], Jianwen Su[1,2], Songyun Deng[1,2], Kun Chen[1,2] and Bin Yu[1,2]

[1] Division of Orthopaedics and Traumatology, Department of Orthopaedics, Nanfang Hospital, Southern Medical University, Guangzhou, Guangdong, China
[2] Guangdong Provincial Key Laboratory of Bone and Cartilage Regenerative Medicine, Nanfang Hospital, Southern Medical University, Guangzhou, Guangdong, China
[3] Department of Orthopaedics, Xiangyang No.1 People's Hospital, Hubei University of Medicine, Xiangyang, Hubei, China

Corresponding author
Bin Yu, yubin@smu.edu.cn

## ABSTRACT

**Background**. Postmenopausal osteoporosis and osteoporosis-related fractures are world-wide serious public health problem. Recent studies demonstrated that inhibiting caveolin-1 leads to osteoclastogenesis suppression and protection against OVX-induced osteoporosis. This study aimed to explore the mechanism of caveolin-1 mediating bone loss and the potential therapeutic target.

**Methods**. Thirty C57BL/6 female mice were allocated randomly into three groups: sham or bilateral ovariectomy (OVX) surgeries were performed for mice and subsequently daidzein or vehicle was administrated to animals (control, OVX + vehicle and OVX + daidzein). After 8-week administration, femurs were harvested for Micro-CT scan, histological staining including H&E, immunohistochemistry, immunofluorescence, TRAP. Bone marrow endothelial cells (BMECs) were cultured and treated with inhibitors of caveolin-1 (daidzein) or EGFR (erlotinib) and then scratch wound healing and ki67 assays were performed. In addition, cells were harvested for western blot and PCR analysis.

**Results**. Micro-CT showed inhibiting caveolin-1with daidzein alleviated OVX-induced osteoporosis and osteogenesis suppression. Further investigations revealed H-type vessels in cancellous bone were decreased in OVX-induced mice, which can be alleviated by daidzein. It was subsequently proved that daidzein improved migration and proliferation of BMECs hence improved H-type vessels formation through inhibiting caveolin-1, which suppressed EGFR/AKT/PI3K signaling in BMECs.

**Conclusions**. This study demonstrated that daidzein alleviates OVX-induced osteoporosis by promoting H-type vessels formation in cancellous bone, which then promotes bone formation. Activating EGFR/AKT/PI3K signaling could be the critical reason.

## INTRODUCTION

Osteoporosis is a systematic skeletal disease characterized by low bone mass and weakened microarchitecture of bone tissue which results a consequent increase in bone fragility and fracture risk (*Compston, McClung & Leslie, 2019*; *Reid, 2020*). One in three women aged 50 or over will have a bone fracture due to osteoporosis, compared with one in five men in their remaining lifetime (*Brown, 2017*). Postmenopausal osteoporosis and osteoporosis-related fractures are world-wide serious public health problem, among which hip and vertebral fractures are associated with increased mortality, up one-quarter of hip fractures die within one year (*Brown, 2017*; *Fink et al., 2019*). Nevertheless, as life expectancy increasing, postmenopausal osteoporosis fosters more need for research.

The occurrence of osteoporosis can be attributed to the unbalance of bone metabolism; both impaired osteogenesis and overactivated osteoclastogenesis. Recently, osteoporosis development has been reported to be concerned with caveolin, a protein mostly locates at caveolae and functions in various physiological processes (*Powter et al., 2015*; *Simon et al., 2020*; *Zhou et al., 2021*). Caveolin-1 (Cav-1), the most studied member in caveolin protein family, is recently identified as a functional factor for regulating osteoclast differentiation (*Hada et al., 2012*; *Huang et al., 2018*; *Zou et al., 2020*). Silencing caveolin-1 results in inhibition of osteoclast formation and protection against ovariectomy (OVX)-induced osteoporosis (*Lee et al., 2015a*; *Lee et al., 2015b*; *Zou et al., 2021*). Studies have shown that the flavonoid daidzein has a wide range of pharmacodynamic properties in the treatment of osteoporosis (*Bellavia et al., 2021*; *Laddha & Kulkarni, 2023*). Interestingly, daidzein has been proved to be a specific inhibitor of caveolin-1 (*Liu et al., 2020*; *Xie et al., 2019*). However, it remains to be seen whether daidzein ameliorates bone loss by inhibiting Cav-1.

That skeletal homeostasis is tightly coupled to growth of blood vessels has been widely studied (*Kusumbe, Ramasamy & Adams, 2014*). Certain capillaries support perivascular osteoprogenitor cells and thereby promote bone formation (*Ramasamy et al., 2016*). Alterations in the skeletal microvasculature might be the critical cause of impaired hematopoiesis and osteogenesis in human subjects with primary osteoporosis (*Kusumbe, Ramasamy & Adams, 2014*). Recently, a specific vessel subtype in bone, which was termed as H-type vessels, has been proved that it can stimulate proliferation and differentiation of osteoprogenitors near the growth plate in the metaphysis and both the periosteum and endosteum of the diaphysis (*Peng et al., 2020*). Furthermore, postmenopausal and age-related osteoporosis is associated with decreases in H-type vessels (*Ramasamy et al., 2014*). Wang and colleagues reported that the abundance of H-type vessels is an important indicator of bone loss in aged human subjects and in those with osteogenesis (*Wang et al., 2017*; *Zhu et al., 2019*). Nevertheless, it remains to be clarified whether Cav-1 may mediate bone metabolism in osteogenesis subjects *via* H-type vessels formation.

## MATERIALS AND METHODS

### Reagents

rat-anti-endomucin (EMCN) (sc-65495; Santa Cruz Biotechnology, Dallax, TX, USA), tartrate resistant acid phosphatase (TRAP) (No:294-67001; Wako, Richmond, VA, USA), rabbit-anti-osteocalcin (OCN) (DF12303; Affinity, Jiangsu, China), erlotinib (HY-50896; MechemExpress, Monmouth Junction, NJ, USA), daidzein (cas#486-66-8; MechemExpress, Monmouth Junction, NJ, USA); rabbit-anti-p-Cav-1 (AF3386; Affinity, Jiangsu, China), rabbit-anti-Cav-1 (cat#16447-1-AP; Proteintech, Wuhan, China), rabbit-anti-p-PI3K (AF3241; Affinity, Jiangsu, China), rabbit-anti-PI3k (cat#20584-1-AP; Proteintech, Wuhan, China), rabbit-anti-p-AKT (AF4418; Affinity, Jiangsu, China), rabbit-anti-AKT (cat#10176-2-AP; Proteintech, Wuhan, China), rabbit-anti-p-epidermal growth factor receptor (EGFR) (cat#ET1606-44; HuaBio, Hangzhou, China), rabbit-anti-EGFR (cat#ET1603-37; HuaBio, Hangzhou, China), rabbit-anti-GAPDH (cat#10494-1-AP; Proteintech, Wuhan, China) antibodies.

### Animals and intervention

Thirty 3-month-old C57BL/6 female mice were obtained from the Experimental Animal Center of the Southern Medical University. Animal care and experiments were approved and conducted in accordance with accepted standards of animal care and use as deemed appropriate by the Nanfang Hospital Animal Ethic Committee to the animal protocol (application No: NFYY-2020-0360). Animals were subjected to the following conditions: normal diet, 20–25 °C room temperature, 50–60% relative humidity, and 12 h light/dark cycle. After acclimatizing for 1 week, mice were allocated randomly into three groups: sham group (sham operation was performed and intragastric (ig) administration with 1% methylcellulose 0.1 ml (vehicle) 3 days later), OVX + vehicle group (OVXV) (bilateral ovariectomy operation was performed and ig administration with 0.1 ml 1% methylcellulose 3 days later), OVX + daidzein group (OVXD) (bilateral ovariectomy operation was performed and ig administration with daidzein at a dose of 25 mg kg$^{-1}$ bodyweight 3 days later). The duration of administration proceeded for 8 weeks (5 days a week). At the end of intervention, femurs were harvested for micro-CT imaging and histology staining.

OVX surgeries were performed as previously described (*Johnson et al., 2010*). Briefly, mice were anesthetized with narcolan which was kept at (125 mg kg$^{-1}$) during surgery. A 1.5 cm incision at midline of the dorsal surface was made, followed by two bilateral incisions through the muscle layer to expose the ovaries. Following ovary removal, the skin incision was closed with sutures, and acetaminophen (50 mg kg$^{-1}$) was administered. The same surgical procedures were performed in the sham group without ovary removal. The duration of administration proceeded for 8 weeks (5 days a week). At the end of intervention, mice were then immediately sacrificed by carbon dioxide inhalation method since multiple parts of bone samples ought to be harvested for further analysis. Femurs and tibias were disarticulated and soft tissue was excised. Left femurs were fixed with 4% paraformaldehyde for 48 h, then scanned with micro-CT. Left tibias were dissected free of soft tissue and stored at −80 °C for mRNA expression analysis. Tibias and femurs from the right side were dissected free of soft tissue and processed for histological analysis.

## Micro-CT for bone mass and microstructure

After anesthesia, animals were sacrificed, left femurs were harvested and fixed overnight in 4% paraformaldehyde, and analyzed by a high-resolution micro-CT ($\mu$CT 80; Scanco Medical, AG, Switzerland). The scan was performed at an isotropic voxel size of 12 $\mu$m, a voltage of 55 kVp, a current of 145 $\mu$A and an integration time of 400 ms. The software CT Analyser (Bruker, Belgium) was used to map the region of interest (ROI) of trabecular bone for analysis of relevant parameters. We analysed the following parameters in the same way as before (*Yao et al., 2021*): bone volume fraction (BV/TV), bone mineral density (BMD), trabecular number (Tb.N), trabecular thickness (Tb.Th), and trabecular separation (Tb. Sp) were calculated.

## HE and TRAP staining

The right femurs were isolated and fixed in 4% paraformaldehyde (pH 7.0) for 48 h, then femurs were decalcified in 10% Ethylene Diamine Tetraacetic Acid (EDTA) (pH 7.2) at 4 °C for 2 weeks. Then, each bone sample was dehydrated in a series of ethanol solutions (75%, 80%, 90%, 95%, 100%), and embedded in paraffin. After cut coronally into 5 $\mu$m sections, tissue slides were prepared for staining with Hematoxylin and Eosin (H&E). Tartrate resistant acid phosphatase (TRAP) staining was performed according to the instructions of the acid phosphatase leukocyte kit (29467001; Wako, Osaka, Japan) and TRAP-positive cells were quantified under a microscope (Nikon Eclipse 80i; Nikon, Tokyo, Japan) to assess osteoclast active. In addition, the average number of osteoclasts was calculated by an independent observer, who was blinded to the groups.

## Immunohistochemical staining/Immunofluorescent staining

Right femurs were collected for analysis of histological changes in response to daidzein intervention. The samples were processed and immunohistochemically stained as per previous methods (*Yao et al., 2021*). The following antibodies were used: Osteocalcin Antibody (1: 150, Affinity, DF12303, OH, USA), Goat anti-Rabbit IgG-HRP Antibody (1: 200, HA1001; HuaBio), DAB kit (ZLI-9018; ZSGB-BIO, Beijing, China). We counted OCN-positive cells per unit length of bone trabeculae in ROI.

For immunofluorescence, sections were deparaffinized after paraffin embedded, antigen retrieved and permeabilized as described above, and then they were subjected to incubation with primary antibody at 4 °C overnight. The following antibodies were used: rabbit-anti-Cav-1 primary antibody (cat#16447-1-AP; Proteintech, Wuhan, China); rabbit-anti-p-EGFR primary antibody (cat#ET1606-44; HuaBio); rat-anti-EMCN primary antibody (V.7C7, Sc-65495, SantaCruz, Dallas, USA); rabbit-anti-CD31 primary antibody (ab222783; Abcam, Cambridge, England); rabbit-anti-Ki67 primary antibody (AF4426; Affinity); goat-anti-rabbit secondary antibody (594, SA00006-4; Proteintech); goat-anti-rat secondary antibody (488, A23240; Abbkine); goat-anti-rabbit secondary antibody (488, Servicebio, GB25303, Wuhan, China); goat-anti-rat secondary antibody (594, HA1112; HuaBio). After washing with PBS, sections were incubated with secondary antibody respectively. We counted positive cells per unit length of bone trabeculae in the ROI by fluorescence microscopy.

 

## Cell wound scratch assay

Rat bone marrow endothelial cells (BMECs) (Seyotin, Guangzhou, China) in logarithmic growth phase were placed in 3.5 cm cell-culture dish at a concentration of $2 \times 10^5$/ml and cultured in a humid cell incubator with 5% CO2 at 37 °C. Two horizontal lines were drawn at the back of 3.5 cm cell-culture dish using marker pens, and 200 µl tips were utilized to draw two vertical lines which were perpendicular to the previous horizontal lines at the bottom of dish. The dish was rinsed with PBS 3 times to eliminate the cells peeled off during wounding. Daidzein was added at concentrations of 0, 10 and 20 µM (the control group had a concentration of 0 µM, and others were experimental groups; n = 3/concentration). BMECs were then cultured in humid incubator with 5% CO2 at 37 °C for 12 and 24 h, respectively. Images were taken at 0, 12, and 24 h after adding drug using an Olympus BX73 (Olympus Corporation, Tokyo, Japan). The migration rate is calculated with reference to previous studies(*Yao et al., 2019*).

## Ki67 cell proliferation assay

To investigate the effect of daidzein and erlotinib (EGFR inhibitor) on endothelial cell, we performed Ki67 staining assay. Rat BMECs were seeded on 12-well culture plate with pre-placed glass coverslips at a density of $4 \times 105$ cells/well (Corning, NY, USA), and were maintained in growth medium (DMEM supplemented with 10% FBS and 100 µg/ml streptomycin and 100 U/ml penicillin). When their confluence rate reached 80–90% after drug intervention (daidzein or erlotinib), cells were incubated in 100% methanol (frozen at −20 °C) for 5 min after rinsed with PBS. Washed with PBS, samples were incubated in PBS (containing 0.1% Triton X-100) for 10 min. Next, washed by PBS, cells were incubated with 1% BSA and 22.52 mg/mL glycine PBST (PBS + 0.1% Tween 20) for 30 min to block the non-specific binding sites of antibodies. In the next step, the method of immuneofluorescence refers to our previous protocol (*Yao et al., 2021*). The following antibodies were used: anti-Ki67 primary antibody (1: 300, ab16667-100; Abcam) and secondary antibody (1: 400, 488, A23240; Abbkine).

## Western Blot assay

Protein samples were subjected to western blot according to our previously described methods (*Yao et al., 2019*). The following antibodies were used: rabbit-anti-p-Cav-1 (AF3386; Affinity), rabbit-anti-Cav-1 (cat#16447-1-AP; Proteintech), rabbit-anti-p-PI3K (AF3241; Affinity), rabbit-anti-PI3K (cat#20584-1-AP; Proteintech), rabbit-anti-p-AKT (AF4418; Affinity), rabbit-anti-AKT (cat#10176-2-AP; Proteintech), rabbit-anti-p-EGFR (cat#ET1606-44; HuaBio), rabbit-anti-EGFR (cat#ET1603-37; HuaBio), rabbit-anti-Angiopoietin1 (AF5184; Affinity) , rabbit-anti- p-VEGFR2 (AF4426; Affinity), rabbit-anti-VEGFR2 (AF4726; Affinity), rabbit-anti-GAPDH (cat#10494-1-AP; Proteintech). Then, the blots were visualised by exposure to X-ray film.

## RT-PCR assay

Total RNA of BMECs was extracted using TRIzol reagent (Invitrogen, Carlsbad, CA, USA) according to manufacturer instructions. Reverse transcription into cDNA was performed using Evo Moloney Murine Leukemia Virus RT Premix (AG11706, Accurate Biology).

Quantitative real-time PCR was performed using SYBR Premix Ex Taq Ⅱ PCR (Takara, Shiga, Japan) on QuantStudio5 (Applied Biosystems, Waltham, MA, USA) according to manufacturer protocol. The Gapdh genes were used as internal control. The relative amount of each gene was calculated using the $2-\Delta\Delta CT$ method. Primer sequences used were as follows: vascular endothelial growth factor (Vegf): (forward 5′-3′, reverse 5′-3′: ACAGAAGGGGAGCAGAAAGC, CTTCATCATTGCAGCAGCCC), (angiopoietin)-1 and Vegf (vascular endothelial growth factor), angiopoietin (Angpt) 1 (forward5′-3′, reverse 5′-3′: ATGCGGTTCAAAACCACACG, TCTGTGAGCTTTCGGGTCTG), Gapdh (forward5′-3′, reverse5′-3′: ACCACAGTCCATGCCATCAC, TCCACCACCCTGTTGCTGTA).

## Statistical analysis

Data in each group were expressed as individual value and means±standard error (SE). To compare the two groups, we used a two-tailed unpaired $t$-test for parametric variables. Comparisons between different groups were conducted using one-way analysis of variance (ANOVA), followed by LSD tests. SPSS v13.0 (SPSS Inc., Chicago, IL, USA) was used for statistical analysis in the study. The level of significance was set at 5%.

# RESULTS

## Cav-1 is involved in OVX-induced osteoporosis and osteogenesis suppression

In order to identify the role of Cav-1 in the occurrence and development of osteoporosis, sham-operation or ovariectomy was performed and bone micro architecture was analysized by micro-CT. We observed OVX-mice showed significantly reduced cancellous bone mass, inducing decreased BV/TV, BMD, Tb.N and Tb.Pf and increased Tb.Sp compared to the sham-operation group (Figs. 1A and 1B). When we intraperitoneally administrated with daidzein, the pharmacological inhibitor of Cav-1, we found it evidently reversed aforementioned changes (BMD: $F = 43.742$, $p < 0.001$; BV/TV: $F = 70.112$, $p < 0.001$; Tb.N: $F = 74.418$, $p < 0.001$; Tb.Pf: $F = 29.115$, $p < 0.001$; Tb.Sp: $F = 19.463$, $p < 0.001$; sham $vs.$ OVXV, $p < 0.001$; OVXV $vs.$ OVXD, $p < 0.001$ in all these microstructure parameters; $n = 6$/group). H&E staining and immunohistochemistry staining for OCN, a bone formation marker, were performed and the results revealed daidzein remarkably alleviated the OVX-induced decrease of osteoblast number (Figs. 1C and 1D) ($F = 40.631$, $p < 0.001$; sham $vs.$ OVXV, $p < 0.001$; OVXV $vs.$ OVXD, $p < 0.001$; $n = 4$ /group) and osteogenic activity (Figs. 1E and 1F) ($F = 38.19$, $p < 0.001$; sham $vs.$ OVXV, $p < 0.001$; OVXV $vs.$ OVXD, $p < 0.001$; $n = 4$ /group) on trabecula bone surface. In addition, we observed the inhibiting effect of daidzein on the OVX-induced increase of osteoclastic activity, according to the TRAP staining (Figs. 1G and 1H) ($F = 11.265$, $p = 0.001$; sham $vs.$ OVXV, $p = 0.004$; OVXV $vs.$ OVXD, $p < 0.001$; $n = 6$/group).

## Daidzein alleviates OVX-induced decreases in H-type vessels in cancellous bone

As accumulating studies argued that the growth of H-type vessels is pivotal to cancellous bone homeostasis (*Kusumbe, Ramasamy & Adams, 2014*; *Liu et al., 2021*; *Ramasamy et al.,*

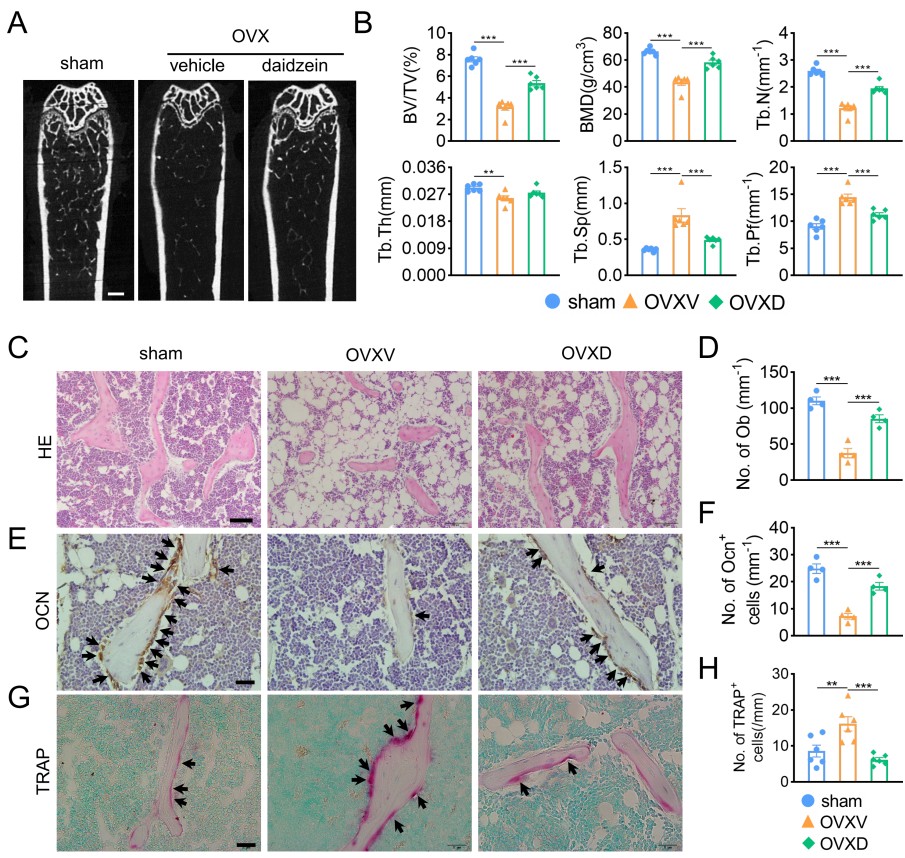

**Figure 1** **Cav-1 is involved in OVX-induced osteoporosis and osteogenesis suppression.** (A) Representative micro-CT images of bone microarchitecture at the femurs from three different groups (control, OVX mice administrated with vehicle or daidzein). (B) Quantitative analysis of trabecular bone parameters, including bone volume fraction (BV/TV), bone mineral density (BMD), trabecular number (Tb.N), trabecular thickness (Tb.Th), trabecular separation (Tb.Sp) and trabecular pattern factor (Tb.Pf) . Scale bar, 0.5 mm. (C, E, G) Representative images of hematoxylin and eosin (HE) staining (C) and quantification of osteoblasts relative to trabecular bone surface (D). Black arrows show osteoblasts on the surface of trabecular bone. Scale bar, 50 μm. Representative images of immunohistochemistry staining for osteocalcin (OCN) (E) and quantification of OCN$^+$ cells on the trabecular bone surface (F) in the femurs of mice. Black arrows show OCN$^+$ cells on the surface of trabecular bone. Scale bar, 20 μm. Representative images of tartrate-resistance acid phosphate (TRAP) staining (G) and quantification of TRAP$^+$ cells relative to trabecular bone surface (H) in femurs. Black arrows show TRAP$^+$ cells. Scale bar, 20 μm. Data were presented as mean ± SE. **$p < 0.01$ and ***$p < 0.001$ as determined by One way ANOVA.

*2014*; *Xie et al., 2014*), we subsequently performed immunofluorescence assays to explore the underlying role of Cav-1 on OVX-induced osteoporosis. The result revealed that Cav-1 expression was upregulated in the OVXV group and daidzein effectively suppressed the Cav-1 expression in cancellous bone (Figs. 2A and 2B) ($F = 28.804$, $p < 0.001$; sham *vs.* OVXV, $p < 0.001$; OVXV *vs.* OVXD, $p < 0.001$; $n = 5$/group). According reported study (*Kusumbe, Ramasamy & Adams, 2014*), EMCN and CD31 were used in this study as specific markers of endothelial cells in H-type vessel. We observed that CD31/EMCN-positive endothelial cells were reduced sharply in the OVXV group, which was effectively alleviated by administration of daidzein (Figs. 2C and 2F) (EMCN: $F = 23.013$, $p < 0.001$; sham

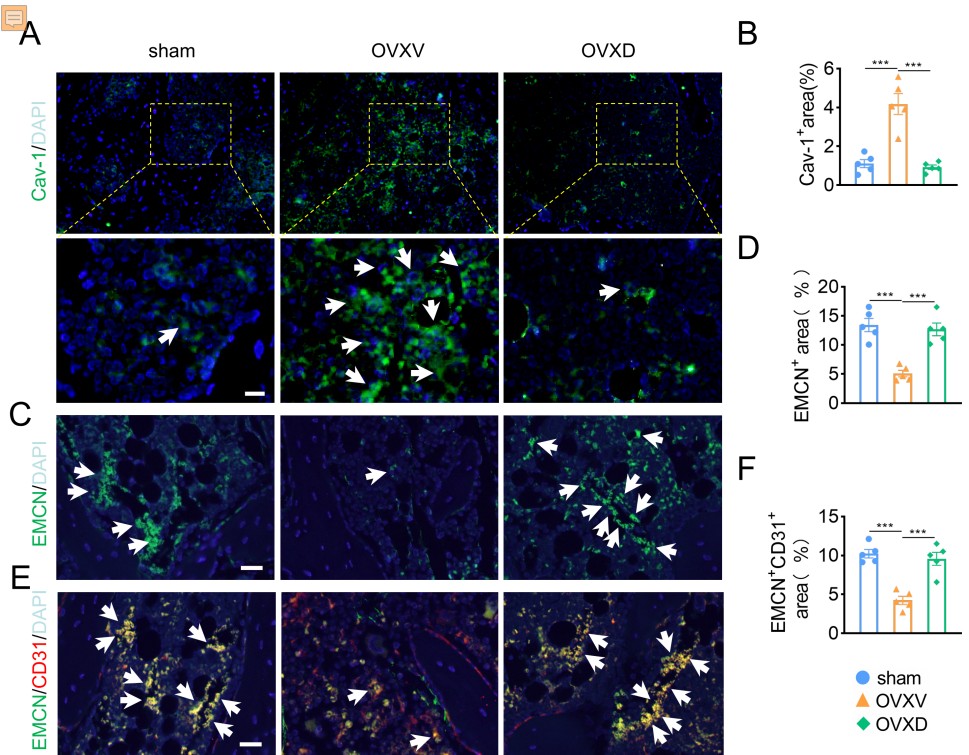

**Figure 2** **Daidzein alleviates OVX-induced decreases in H-type vessels in cancellous bone.** Immunofluorescence staining for Cav-1 (Cav-1) (A), EMCN (C) and double staining for EMCN and CD31 (E) in mice femoral sections in three different groups (control, OVX mice administrated with vehicle or daidzein) and white arrows indicate Cav-1$^+$ cells, EMCN$^+$ cells and EMCN$^+$/CD31$^+$ cells respectively in A, C and E. Scale bar, 20 µm. (B, D and F) Quantitative analysis of Cav-1$^+$ cells, EMCN$^+$ cells and EMCN$^+$/CD31$^+$ cells respectively per area of trabecular bone surface. Data were presented as mean ± SE . ***$p < 0.001$ as determined by one-way ANOVA.

*vs.* OVXV, $p < 0.001$; OVXV *vs.* OVXD, $p < 0.001$; CD31/EMCN: $F = 25.739$, $p < 0.001$; sham *vs.* OVXV, $p < 0.001$; OVXV *vs.* OVXD, $p < 0.001$; $n = 5$/group).

## Daidzein improves migration and proliferation of BMECs

In order to explore the underlying mechanism of daidzein promoting H-type vessel formation, we performed wound scratch assay with BMECs. The results revealed inhibiting Cav-1 by daidzein could promote migration of BMECs (Figs. 3A and 3B) ($F = 7.56$, $p = 0.003$; 0 *vs.* 10 µm, $p < 0.014$; 0 *vs.* 20 µm, $p = 0.001$; $n = 6$ /group). Moreover, Ki67 assay was performed and we found daidzein also promoted proliferation of BMECs (Figs. 3C and 3D) (0 *vs.* 10 µm in 24 h, $t = -3.055$, $p = 0.022$, $n = 4$ /group). Two critical biomarkers for angiogenesis, *Angpt-1* and *Vegf*, were found upregulated in daidzein group, according the PCR assay (Figs. 3E and 3F) (*Angpt-1*: $t = -6.818$, $p = 0.002$; *Vegf*: $t = -3.369$, $p = 0.028$; $n = 3$ /group). Furthermore, we performed western blotting and found ANGPT-1 and phospho-VEGFR in BMECs were upregulated after the administration with 10 µM daidzein (Figs. 3G and 3H) (ANGPT-1: $t = 6.812$, $p = 0.029$; p-VEGFR: $t = 8.444$, $p = 0.018$; $n = 3$/group).

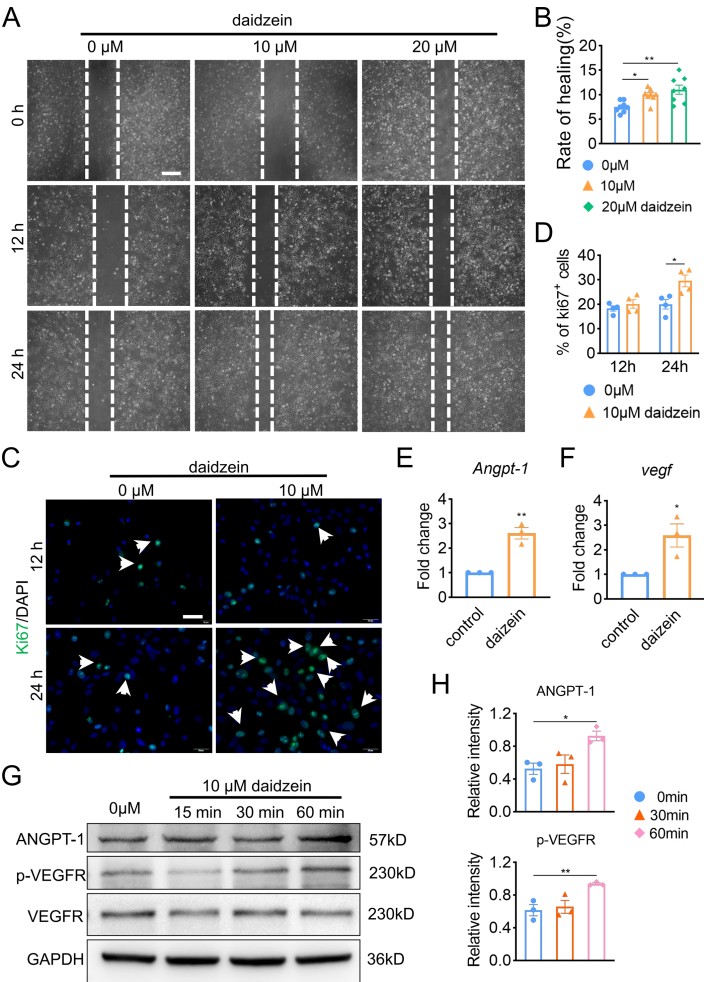

**Figure 3** **Daidzein improves migration and proliferation of BMECs.** (A) BMECs were cultured and treated with different dose of daidzein (0, 10, 20 μM). Scratch wound healing assay was performed to observe the effect of daidzein on the migration of BMECs. Scale bar, 100 μm. (B) Quantitative analysis of wound healing rate. (C) Representative images of immunofluorescence staining for Ki67 (green) in BMECs treated with different dose of daidzein (0, 10 μM). Nuclei were stained with DAPI (blue). White arrows indicate Ki67+ cells. Scale bar, 100 μm. (D) Quantification of the percentage of Ki67+ nuclei among the total nuclei. (E, F) RNA was isolated from BMECs in the control and daidzein groups, and the mRNA expression of *Angpt-1* and *Vegf* was detected with qRT-PCR. (G) BMECs were cultured and treated with different dose of daidzein, protein was harvested at indicated time points, western blotting was performed to detect the expression and ANGPT-1 phospho-VEGFR and GAPDH was used as reference proteins for data normalization. (H) ANGPT-1 and phosphorylation levels of VEGFR were quantified and normalized to total VEGFR. Data were presented as mean ± SE. *$p < 0.05$ and **$p < 0.01$ as determined by One way ANOVA or two-tailed unpaired $t$-test.

## Daidzein improves H-type vessels formation through the AKT/EGFR signaling pathway

The underlying mechanism of the improved migration and proliferation of BMECs by daidzein remains to be established. Reported literatures were reviewed and we concluded the PI3K/AKT and EGFR signaling was observed mostly in study on migration and

proliferation of endothelial cells (*Bouvard et al., 2015*; *Gu et al., 2009*; *Murphrey, Quaim & Varacallo, 2022*; *Zhang et al., 2018*). On this context, we performed western blotting and found phosphorylation of AKT, EGFR and PI3Kin BMECs were upregulated after the administration with 5 μM or 10 μM daidzein (Figs. 4A and 4D) (AKT: $F = 7.131$, $p = 0.026$; 0 *vs.* 60 min, $p = 0.009$; EGFR: $F = 14.114$, $p = 0.005$; 0 *vs.* 30 min, $p = 0.009$; 0 *vs.* 60 min, $p = 0.002$; PI3K: $F = 7.971$, $p = 0.02$; 0 *vs.* 30 min, $p = 0.032$; 0 *vs.* 60 min, $p = 0.008$; $n = 3$/group). Immunofluorescence results confirmed daidzein suppressed the OVX-induced elevation of Cav-1 in endothelial cells in femoral cancellous bone (Figs. 4E and 4F) ($F = 50.145$, $p < 0.001$; sham *vs.* OVXV, $p < 0.001$; OVXV *vs.* OVXD, $p = 0.001$; $n = 5$/group). We also observed OVX suppressed activation of EGFR in endothelial cells while daidzein attenuated this effect (Figs. 4G and 4H) ($F = 28.541$, $p < 0.001$; sham *vs.* OVXV, $p < 0.015$; OVXV *vs.* OVXD, $p < 0.001$; $n = 5$/group). These results demonstrated daidzein improved proliferation and migration of BMECs through the AKT/PI3K and EGFR signaling pathways.

## Cav-1 inhibits migration and proliferation of BMECs through the suppressing EGFR/AKT/PI3K signaling

Subsequently, in order to fulfill the episode of daidzein promoting migration and proliferation of endothelial cells, we performed wound scratch assay and Ki67 assay and found suppressing EGFR with erlotinib improved BMECs migration (Figs. 5A and 5B) ($F = 11.174$, $p = 0.009$; 0 *vs.* 10 μm, $p = 0.003$; $n = 3$/group) and proliferation (Figs. 5C and 5D) (0 *vs.* 10 μm in 12 h, $t = 6.259$, $p = 0.001$; 0 *vs.* 10 μm in 24 h, $t = 3.938$, $p = 0.008$; $n = 4$/group) We then performed western blot to explore the relationship between EGFR and AKT/PI3K. The results revealed inhibiting EGFR upregulated the activity of AKT/PI3K signaling (Figs. 5E and 5F) (AKT: $F = 7.716$, $p = 0.022$; 0 *vs.* 4 h, $p = 0.02$; 0 *vs.* 8 h, $p = 0.011$; PI3K: $F = 6.763$, $p = 0.029$; 0 *vs.* 8 h, $p = 0.012$; $n = 3$/group), while the expression of Cav-1 remained unchanged ($F = 3.33$, $p = 0.106$, $n = 3$/group). Further, immunofluorescence staining for EMCN/Ki67 showed that Cav-1 inhibited the proliferation of EMCN-positive endothelial cells in the bone marrow ($F = 14.249$, $p = 0.002$, $n = 4$/group). These results proved that Cav-1 ought to be the upstream of EGFR/AKT/PI3K signaling in regulating migration and proliferation of endothelial cells.

## DISCUSSION

Several recent investigations had reported that the prevalence of osteoporosis increases with age and differs by race, and it is a major health problem that affects over 200 million people worldwide (*Cheng, Wentworth & Shoback, 2020*). Osteoporotic fracture, the main complication of osteoporosis, is frequently associated with chronic pain, disability, decreased quality of life and 21% to 30% of patients who suffer hip fractures die in a year (*Jin, 2018*). One treatment for osteoporosis is hormone replacement therapy, studies have shown that phytoestrogens like daidzein exhibit protective effects on bone loss (*Akhlaghi et al., 2020*; *Kim, Kim & Yang, 2021*; *Mansoori et al., 2016*). Bone loss induced by estrogen deficiency is a complex effect of a congregation of pathways and cytokines which work in a cooperative fashion to regulate processes like osteoblastogenesis and

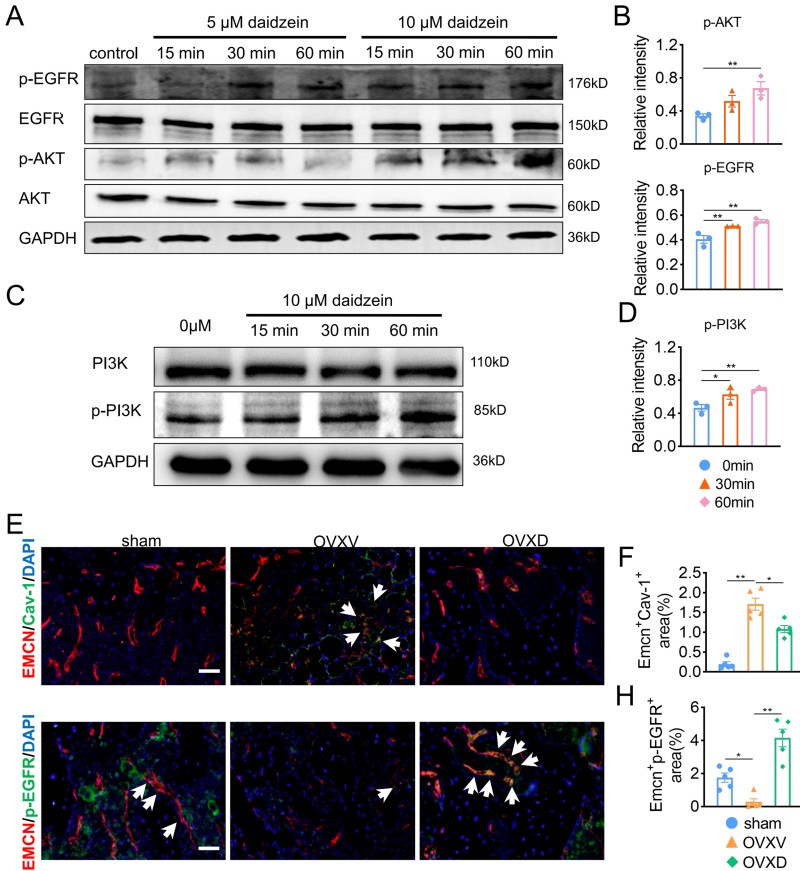

**Figure 4  Daidzein improves H-type vessels formation through the AKT/EGFR signaling pathway.** (A and C) BMECs were cultured and treated with different dose of daidzein, protein was harvested at indicated time points, western blotting was performed to detect the expression and phosphorylation of AKT, EGFR, PI3K and GAPDH was used as reference proteins for data normalization. (B and D) The phosphorylation levels of aforementioned protein were quantified and normalized to total EGFR, AKT and PI3K respectively. (E and F) Representative images of double immunofluorescence staining for EMCN /Cav-1 (E) and EMCN/p-EGFR (G). White arrows indicate EMCN[+]/Cav-1[+] and EMCN[+]/p-EGFR[+] cells. Scale bar, 50 μm. (F and H) Quantitative analysis of EMCN[+]/Cav-1[+] cells and EMCN[+]/p-EGFR[+] cells respectively per area of trabecular bone surface. Data were presented as mean ± SE. $*p < 0.05$, $**p < 0.01$ and $***p < 0.001$ as determined by one-way ANOVA.

osteoclastogenesis (*Faienza et al., 2013*). In order to confirmed the anti-osteoporosis effect of daidzein and explore the underlying mechanism, we launched the investigation. In this study, we creatively found the Cav-1 inhibitor; daidzein alleviates OVX-induced bone loss through preventing EGFR/AKT/PI3K signaling inactivation in vascular endothelial cells and consequent promoting in H-type vessels in cancellous bone. This finding suggests a potential therapy for osteoporosis.

According to Micro-CT results in this study, we found an increase of trabecular bone mass after OVX mice treating with daidzein compared with the OVXV group, which is identical with several studies (*Maes et al., 2010*; *Powter et al., 2015*; *Ramasamy et al., 2014*). Daidzein demonstrated particular therapeutic effect on cancellous bone, which

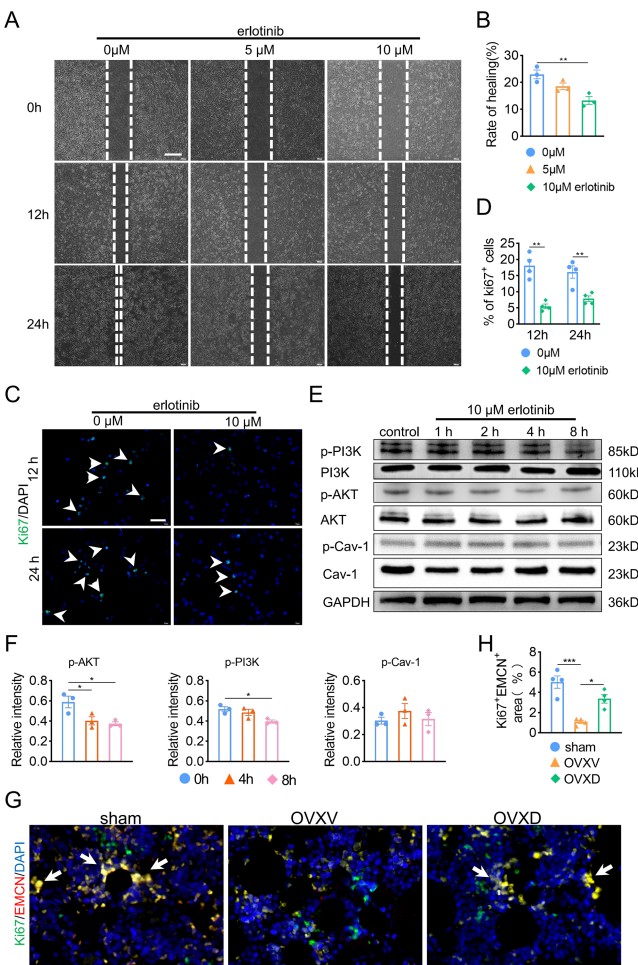

**Figure 5** **Cav-1 inhibits migration and proliferation of BMECs through suppressing EGFR/AKT/PI3K signaling.** (A) BMECs were cultured and treated with different dose of erlotinib (0, 5, 10 μM). Scratch wound healing assay was performed to observe the effect of erlotinib on the migration of BMECs. Scale bar, 100 μm. (B) Quantitative analysis of wound healing rate. (C) Representative images of immunofluorescence staining for Ki67 (green) in BMECs treated with different dose of erlotinib (0, 10 μM). Nuclei were stained with DAPI (blue). White arrows indicate Ki67[+] cells. Scale bar, 100 μm. (D) Quantification of the percentage of Ki67[+] nuclei among the total nuclei. (E) Representative western blot images for p-AKT, AKT, p-Cav-1, Cav-1, PI3K, and p-PI3K in the BMECs treated with 10 μM erlotinib and GAPDH was used as reference proteins for data normalization. (F) The phosphorylation levels of AKT, Cav-1, PI3K protein were quantified and normalized to their total proteins respectively. (G) Representative images of double immunofluorescence staining for Ki67/EMCN. White arrows indicate Ki67[+]/EMCN[+] cells. Scale bar, 50 μm. (H) Quantitative analysis of Ki67[+]/EMCN[+] cells respectively per area of trabecular bone surface. Data were presented as mean ± SE. *$p < 0.05$ and **$p < 0.01$ as determined by one-way ANOVA or two-tailed unpaired $t$-test.

in cortical bone showed scarcely changed yet (*Fonseca & Ward, 2004*), suggesting that daidzein expressed mostly in cancellous bone and exhibited anti-osteoporosis effect on myeloid-derived cells most.

Cav-1 is an essential structural constituent of caveolae implicated in cellular mitogenic signaling, angiogenesis and senescence (*Ramasamy et al., 2016*). *Rippe et al. (2012)* reported

both mRNA and protein expression of Cav-1 increased in induced senescent endothelial cells, which were stained positively with SA-β-gal. In addition, it has been shown that replicative senescent cells re-enter the cell cycle upon EGF stimulation after down-regulation of Cav-1 (*Schreier, Gekle & Grossmann, 2014*). *Kong et al. (2018)* showed decreased Cav-1 affects not only the cell morphology and size, but also some important function of endothelial cells like migration and angiogenesis. Together, these studies verified the crucial role of Cav-1 in signal transduction and leading to cellular senescence. In our immunofluorescence assays, we observed Cav-1 was significantly up-regulated in endothelial cells after mice were ovariectomized. We hypothesized that Cav-1 mediating senescence of endothelial cells could be an interpretation for impaired H-type vessel formation and the resultant osteoporosis. Further researches are needed to confirm.

The complex heterogeneous micro-environment of skeletal system is dependent upon the strict coordination of multiple cell types (*e.g.*, osteoblasts, osteoclast, endothelial cells) to sustain the constant remodeling process of bone formation and bone resorption over a lifetime but also to redirect cell populations during periods of bone repair and regeneration (*Park et al., 2017*). In addition to historic studies highlighting the close proximity of vascular and osteoblastic cells, potential roles of angiogenic blood vessel growth in fracture healing have been proposed (*Mizuta et al., 2020*). A growing body of evidence, both in clinical and experimental, demonstrates the crucial role of angiogenesis in bone hemostasis, and the coupling of angiogenesis-osteogenesis is verified (*Gu et al., 2009*; *Jin, 2018*; *Quan et al., 2019*). In this study, we speculated skeletal vasculature may have a link to osteoporosis and its alleviation. Then we performed immunofluorescent staining of EMCN and colocalization of EMCN with Cav-1 and p-EGFR. The result of EMCN staining revealed a decreased expression of EMCN in the OVXV group compared with sham group. Concurrently, its restoration was observed after daidzein administration in the OVXD group. Previous studies have shown that daidzein can bind to $\alpha$ and $\beta$ estrogen receptors, which cause estrogen like effects and thus is able to counteract the discrepancy between bone growth and resorption in osteoporotic bone (*Setchell & Lydeking-Olsen, 2003*). Another research argued daidzein could bind receptor activator of NF-$\kappa$B ligand (RANKL) and support osteoprotegerin (OPG) function to prevent bone resorption, as well as regulate osteogenic differentiation markers (*Zaklos-Szyda et al., 2020*). In our study, we found daidzein can alleviate osteoporosis in OVX mice through promoting angiogenesis. We speculated OVX-induced osteoporosis may be caused by decreased intramedullary vascular number, H-type vessel particularly, and the number of blood vessels and bone volume in mice increased when treated with daidzein. Therefore, we concluded that daidzein ameliorates osteoporosis development through up-regulating intramedullary blood vessels formation in cancellous bone.

The activated EGFR can bind to several signaling proteins and stimulate the activation of many signaling pathways involving in cellular proliferation, differentiation, migration and apoptosis (*Liu et al., 2018*). EGFR signaling promotes endochondral ossification process which regulates long bone development, and also has a great effect on angiogenesis of multi-type cells (*Cheng et al., 2021*).

Several researches sentence that EGFR has demonstrated expression on vascular endothelial cell and binding by various ligands results in routine cellular processes such as proliferation, differentiation and cellular development (*Jiang et al., 2000*). In addition, the EGFR is a versatile signaling pathway integrator associated with vascular homeostasis, modulating basal vascular tone and tissue homeostasis (*Bouvard et al., 2015*). According to the colocalization staining of EMCN and Cav-1, we observed daidzein rescued OVX-induced osteoporosis by inhibiting Cav-1 and promoting blood vessels formation. Next, we conducted colocalization of EMCN with p-EGFR staining, and observed a decrease of EMCN as well as p-EGFR in the OVXV group than in the sham group, meanwhile; we discovered an up-regulation of EMCN and p-EGFR after daidzein administration in the OVXD group compared with the OVXV group. In conclusion, we argued the expression of EMCN is probably modulated by EGFR, and they show a positive correlation manner in our assays.

In order to elucidate the bone-sparing mechanism of daidzein, we performed OCN immunohistochemical staining and TRAP staining. The results demonstrated a prominent decrease of OCN$^+$cells as well as an increase of TARP$^+$cells in the OVXV group, and it was reversed by treating with daidzein, showing a significant increase of OCN$^+$cells as well as a decrease of TARP$^+$cells in the OVXD group, which represented the promoting effect of daidzein on osteoblasts and inhibiting effect on osteoclasts. Consequently, our findings are in line with a selection of articles that report the promoting effects of daidzein on osteoblasts and inhibiting effects on osteoclasts (*Akhlaghi et al., 2020*; *Faienza et al., 2013*; *Fonseca & Ward, 2004*; *Mansoori et al., 2016*; *Ramasamy et al., 2014*)

It was reported that the PI3K/AKT signaling pathway has crucial roles in vascular endothelial cell proliferation and migration (*Carmeliet & Jain, 2011*; *Han et al., 2015*; *Mazzieri et al., 2011*; *Semenza, 2007*). Hence, constitutively active PI3K/AKT induced angiogenesis and that inhibition of PI3K/AKT signaling interfered with angiogenesis (*Bouvard et al., 2015*; *Jiang et al., 2000*). In western blot assay, we demonstrated a decrease of Cav-1 as well as an increase of p-EGFR and p-AKT concurrently.

Angiogenesis is a physiological or pathological process from which new blood vessels develop from pre-existing vessels, and in which ECs is the key (*Semenza, 2007*). During this process, one of the most potent pro-angiogenic factors is VEGF and its receptors VEGFR (*Carmeliet & Jain, 2011*). In addition to VEGF, other angiogenic factors including FGF, HGF, TSP-1, endostatin and phospholipids such as lysophosphatidic acid all act on ECs directly or indirectly by inducing the expression of angiogenic factors (*Fonseca & Ward, 2004*). Apart from VEGF, ANGPT1 and ANGPT2, which bind to receptor tyrosine kinase Tie-2, are also important factors in the process of regulating and controlling angiogenesis (*Mazzieri et al., 2011*). ANGPT1 is fundamental to physiological angiogenesis, including endothelial cell survival, vascular branching, and pericyte recruitment. Increasing amounts of experimental data have suggested that ANGPT1 contributes to the stabilization of newly organized blood vessels (*Han et al., 2015*). Hence, we focused on detecting VEGF and ANGPT1 genes and found daidzein dramatically up-regulated the expression of both markers and promoted H-type vessels formation. We also found that ANGPT1 and

Phospho-VEGFR protein levels in BMEC were upregulated by daidzein. We uncovered another interpretation for daidzein ameliorating osteoporosis by inhibiting Cav-1.

The results of this study should be interpreted in the context of its potential limitations. Firstly, we did not verify the effect of EGFR/AKT/PI3K signaling on Cav-1-induced bone loss *in vivo*, while it is certain that daidzein promotes endothelial cells proliferation and migration. Secondly, we observed both promoting effect of daidzein on osteoblasts and inhibiting effect on osteoclasts. Further investigation is needed to uncover the mechanism of daidzein alleviating osteoporosis in terms of osteoclastogenesis.

## CONCLUSIONS

In this study, we discovered a novel anti-osteoporosis effect of daidzein in terms of coupling of angiogenesis and osteogenesis. Inhibiting Cav-1, daidzein activates EGFR/PI3K/AKT signaling in endothelial cells and hence promotes H-type vessel formation in cancellous bone, which is the critical mechanism of its protection from OVX-induced bone loss. In this case, we expect Cav-1/EGFR signaling pathway may be another potential anti-osteoporosis therapeutic target.

## ACKNOWLEDGEMENTS

We are grateful to Professor Xianrong Zhang for her careful review of the raw data in this study and for her suggestions on improving this manuscript.

### Funding

This study was supported by the National Natural Science Foundation of China (81830079), the Guangdong Basic and Applied Research Foundation (Grant Number: 2019A1515110818) and the President Foundation of Nanfang Hospital, Southern Medical University (2019Z012). The funders had no role in study design, data collection and analysis, decision to publish, or preparation of the manuscript.

### Grant Disclosures

The following grant information was disclosed by the authors:
The National Natural Science Foundation of China: 81830079.
Guangdong Basic and Applied Research Foundation: 2019A1515110818.
President Foundation of Nanfang Hospital, Southern Medical University: 2019Z012.

### Competing Interests

The authors declare there are no competing interests.

### Author Contributions

- Junjie Jia performed the experiments, analyzed the data, prepared figures and/or tables, authored or reviewed drafts of the article, and approved the final draft.

- Ruiyi He performed the experiments, analyzed the data, prepared figures and/or tables, and approved the final draft.
- Zilong Yao analyzed the data, prepared figures and/or tables, authored or reviewed drafts of the article, and approved the final draft.
- Jianwen Su performed the experiments, prepared figures and/or tables, and approved the final draft.
- Songyun Deng performed the experiments, prepared figures and/or tables, and approved the final draft.
- Kun Chen performed the experiments, prepared figures and/or tables, and approved the final draft.
- Bin Yu conceived and designed the experiments, authored or reviewed drafts of the article, and approved the final draft.

## Animal Ethics

The following information was supplied relating to ethical approvals (i.e., approving body and any reference numbers):

Nanfang Hospital Animal Ethic Committee (application No: NFYY-2020-0360).

## Data Availability

The raw measurements are available in Supplementary File 1.

## Supplemental Information

Supplemental information for this article can be found online at http://dx.doi.org/10.7717/peerj.16121#supplemental-information.

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
