# Peer review of "Daidzein alleviates osteoporosis by promoting osteogenesis and angiogenesis coupling"

_PeerJ, doi:10.7717/peerj.16121_

## Round 0.1 · original submission · Major Revisions

Please address the concerns of both reviewers and amend the manuscript accordingly.

Reviewer 1 ·

Basic reporting

Remarks to the Author:
The article “Daidzein alleviates osteoporosis by promoting osteogenesis and angiogenesis coupling’’ is an interesting study detecting the anti-osteoporosis effect of diadzein by enhancing bone formation causing.
However, the existing data are partially supportive of the authors' conclusions. Additional experiments are needed to fully support their findings. Major revision is needed in recommending this manuscript for publication.

Major Issue:
1. Authors have mentioned that Cav-1 inhibited the migration and proliferation of bone marrow cells by downregulating the EGFR/AKT/PI3K signaling pathway. However, they have shown these in only one cell line hence invivo model is essential to further confirm this conclusion.

Minor issues:
1. Authors mentioned that diadzein enhances bone cells migration. However authors have only showed the genetic expression of VEGF and Angpt1 in rat bone marrow endothelial cells. Authors should include the protein level of these molecules along with VEGFR.

Experimental design

no comment

Validity of the findings

no comment

Additional comments

no comment

Annotated reviews are not available for download in order to protect the identity of reviewers who chose to remain anonymous.

Reviewer 2 ·

Basic reporting

This manuscript is interesting and well done, however I propose some suggestions. In particular:
- The title indicates “Daidzein alleviates osteoporosis by promoting osteogenesis and angiogenesis coupling”, but no indication of this isoflavone and its activities on osteoporosis in general and caveolin-1 in particular, are indicated in the introduction. Please insert these information (see Bellavia et al. Flavonoids in Bone Erosive Diseases: Perspectives in Osteoporosis Treatment. Trends in Endocrinology & Metabolism, February 2021, Vol. 32, No. 2 https://doi.org/10.1016/j.tem.2020.11.007 ).
- I suggest changing/removing the colours in the graphs where there is no colour legend, as it can lead to confusion for the reader. In particular, in Figure 5 B and F;

Experimental design

the study is experimentally correct and well done.

Validity of the findings

the study is not very innovative, in fact, the effect of daidzein on caveolin, its effect on endothelial cells and its effect on osteoporosis are known. This work brings it all together.

---

## Round 0.2 · Minor Revisions

Please address remaiing concerns of the reviewer and revise your manuscript accordingly.

**Language Note:** The review process has identified that the English language must be improved. PeerJ can provide language editing services - please contact us at copyediting@peerj.com for pricing (be sure to provide your manuscript number and title). Alternatively, you should make your own arrangements to improve the language quality and provide details in your response letter. – PeerJ Staff

Reviewer 2 ·

Basic reporting

This manuscript is interesting and well written, and the authors have responded adequately to my requests. However, I suggest that the authors to review the text because there are some mistakes. In particular:
- Line 67: change “homeostas” with “homeostasis”;
- Line 63: insert a space between the word “osteoporosis” and the references;
- Line 64: insert a space between the fullstop and the next sentence;
- Line 65: insert a space between the fullstop and the next sentence;
- Line 97: insert a space between the word “protocol” and the parenthesis;
- Line 247: change “goup” with “group”;
- Line 251: Change "effectly” with “effectively”;
- Line 318: Change "traubecular” with “trabecular”;
- Line 321: Change "therapic” with “therapeutic”;
- Line 331: Change "agiogenesis” with “angiogenesis”;
- Line 343: Change "experimential” with “experimental”;
- Line 348: Change "compaired” with “compared”;
- Line 359: Change "conclued” with “concluded that”;
- Line 367 and 368: Change " celluar” with “cellular”;
- Line 421: Change " therapeautic” with “therapeutic”
It is possible that some errors may have escaped me. I suggest a revision of the manuscript.

Experimental design

experimental Design is correct and well done

Validity of the findings

the obtained results are clear and in line with the described conclusions and the data reported in literature.

Additional comments

none

---

## Round 0.3 · accepted · Accept

Thank you for addressing the remaining issues pointed out by the reviewer and revising the manuscript accordingly. I am pleased to inform you that the revised version is acceptable now.